# Health systems' capacity in availability of human resource for health towards implementation of Universal Health Coverage in Kenya

Ismail Adow Ahmed[1]*, James Kariuki[1], David Mathu[1], Stephen Onteri[1], Antony Macharia[1], Judy Mwai[1], Priscah Otambo[1], Violet Wanjihia[1], Joseph Mutai[1], Sharon Mokua[1], Lilian Nyandieka[1], Elizabeth Echoka[1], Doris Njomo[2], Zipporah Bukania[1]

**1** Centre for Public Health Research, Kenya Medical Research Institute Kenya, Nairobi, Kenya, **2** Kenya Medical Research Institute Kenya, Eastern and Southern Africa Centre of International Parasite Control, Nairobi, Kenya

* ismailadow@yahoo.com

**Data Availability Statement:** The data used to support the findings of this study are included

## Abstract

### Introduction

Kenya faces significant challenges related to health worker shortages, low retention rates, and the equitable distribution of Human Resource for Health (HRH). The Ministry of Health (MOH) in Kenya has established HRH norms and standards that define the minimum requirements for healthcare providers and infrastructure at various levels of the health system. The study assessed on the progress of Universal Health Coverage (UHC) piloting on Human Resource for Health in the country.

### Methods

The study utilized a Convergent-Parallel-Mixed-Methods design, incorporating both quantitative and qualitative approaches. The study sampled diverse population groups and randomly selected health facilities. Four UHC pilot counties are paired with two non-UHC pilot counties, one neighboring county and the second county with a geographically distant and does not share a border with any UHC pilot counties. Stratification based on ownership and level was performed, and the required number of facilities per stratum was determined using the square root allocation method. Data on the availability of human resources for health was collected using a customized Kenya Service Availability and Readiness Assessment Mapping (SARAM) tool facilitated by KoBo ToolKit™ open-source software. Data quality checks and validation were conducted, and the HRH general service availability index was measured on availability of Nurses, Clinician, Nutritionist, Laboratory technologist and Pharmacist which is a minimum requirement across all levels of health facilities. Statistical analyses were performed using IBM SPSS version 27 and comparisons between UHC pilot counties and non-UHC counties where significance threshold was established at $p < 0.05$. Qualitative data collected using focus group discussions and in-depth interview guides.

within the main article. There is no need for Supporting Information files.

**Funding:** The research leading to these results has received funding from the Government of Kenya through the Board of Management and the Office of the Director General, Kenya Medical Research Institute, Kenya. The funders had no role in study design, data collection and analysis, decision to publish, or preparation of the manuscript.

**Competing interests:** The authors have declared that no competing interests exist.

Ethical approval and research permits were obtained, and written informed consent was obtained from all participants.

## Results

The study assessed 746 health facilities with a response rate of 94.3%. Public health facilities accounted for 75% of the sample. The overall healthcare professional availability index score was 17.2%. There was no significant difference in health workers' availability between UHC pilot counties and non-UHC pilot counties at P = 0.834. Public health facilities had a lower index score of 14.7% compared to non-public facilities at 27.0%. Rural areas had the highest staffing shortages, with only 11.1% meeting staffing norms, compared to 31.8% in urban areas and 30.4% in peri-urban areas. Availability of health workers increased with the advancement of The Kenya Essential Package for Health (KEPH Level), with all Level 2 facilities across counties failing to meet MOH staffing norms (0.0%) except Taita Taveta at 8.3%. Among specific cadres, nursing had the highest availability index at 93.2%, followed by clinical officers at 52.3% and laboratory professionals at 55.2%. The least available professions were nutritionists at 21.6% and pharmacist personnel at 33.0%. This result is corroborated by qualitative verbatim.

## Conclusion

The study findings highlight crucial challenges in healthcare professional availability and distribution in Kenya. The UHC pilot program has not effectively enhanced healthcare facilities to meet the standards for staffing, calling for additional interventions. Rural areas face a pronounced shortage of healthcare workers, necessitating efforts to attract and retain professionals in these regions. Public facilities have lower availability compared to private facilities, raising concerns about accessibility and quality of care provided. Primary healthcare facilities have lower availability than secondary facilities, emphasizing the need to address shortages at the community level. Disparities in the availability of different healthcare cadres must be addressed to meet diverse healthcare needs. Overall, comprehensive interventions are urgently needed to improve access to quality healthcare services and address workforce challenges.

## 1 Introduction

The focus on Universal Health Coverage (UHC) has gained momentum in the recent years where World Health Assembly and the United Nations General Assembly has called on countries to accelerate transition towards universal access to affordable and quality healthcare services [1]. Kenya adopted UHC as one of the big four priority agenda by His Excellency the President, with an aspiration that by 2022, all persons in Kenya will be able to use the essential services they need for their health and wellbeing through a single unified benefit package, without the risk of financial catastrophe [2]. Achieving universal health coverage requires continuing political commitment and leadership to distribute available resources, especially human resources for health (HRH), in an efficient, equitable and sustainable manner to match population needs [3].

The health workforce is the cornerstone of every health system and critical to the provision of quality health services, improving population health, ensuring universal health coverage and the achievement of the Sustainable Development Goals. The Global Strategy on human resources for health: Workforce 2030 emphasizes that health systems can only function well when they have sufficient well-trained, competent, responsive, motivated, productive and equitably distributed health staff. In 2017, African region adapted 2012–2015 road map for scaling up HRH for improved health service delivery at the sixty-second session of the regional committee [4]. Despite having 25% of the world's illness burden, WHO estimates that Sub-Saharan Africa has only 1.3 per cent of the world's trained health personnel [4]. Africa had an average of 1.30 health workers per 1000 people in 2015, significantly less than the 4.5 needed to meet the Sustainable Development Goals (SDGs). Though gains have been made, Africa has the most severe health personnel shortfall, predicted to reach 6.1 million by 2030, out of the anticipated global health staff need of 14.5 million required for UHC and SDGs [5].

Kenya currently faces significant challenges in overcoming health worker shortages and low retention, as well as difficulty in attaining equitable distribution of human resources for health, particularly in hard-to-reach areas [6]. In Kenya, the total number of the health workers currently employed in the County Departments of Health as well as in the public, faith-based organization (FBO), and private-for-profit health facilities is estimated at 31, 412, these numbers are below the required of 138, 266 healthcare workers as per the Norms and Standards Guidelines by the Ministry of Health [7]. Kenya is one of the countries that has committed to achieving UHC as part of its national development agenda. However, the availability and distribution of human resources for health remains a critical challenge in the country. This study's assessment of the health systems' capacity in the availability of human resources for health in Kenya fills the existing knowledge gap, identifies HRH shortages and distribution disparities, informs policy and planning, and contributes to global knowledge especially low- and middle-income countries. The findings are instrumental in strengthening the health workforce and facilitating the successful implementation of UHC in Kenya. The study assessed the availability and distribution of human resources for health in Kenya's health system in realization of Universal Health Coverage.

## 2. Methodology

### 2.1 Study design

The survey used a convergent parallel mixed methods study design utilizing both qualitative and quantitative methods concurrently. The sample design was structured based on facility ownership and the Kenya Essential Package for Health (KEPH) level. The KEPH service levels are organized from level 1 which is at the community level; level 2 –dispensaries and clinics; level 3 -health centres and maternity homes and sub district hospitals; level 4 -primary facilities which include district hospitals; level 5—secondary facilities/ provincial hospitals to level 6. Data collected between February 2020-March 2020 and October 2020-December 2020. Convergent parallel mixed methods study design demonstrated in Fig 1.

### 2.2 Study sites

It encompasses two groups of counties: UHC pilot Counties (Nyeri, Machakos, Isiolo, and Kisumu) and Non-UHC pilot Counties (Nyandarua, Meru, Kitui, Homabay, Bomet, West Pokot, Taita Taveta, and Bungoma). Government of Kenya launched UHC and piloted in four Counties out of forty-seven. The objective of the study is to assess the readiness and performance of UHC rollout in Kenya. To achieve this, each UHC pilot county is paired with two non-UHC pilot counties. One of these pairs comprises a neighboring county to the UHC pilot

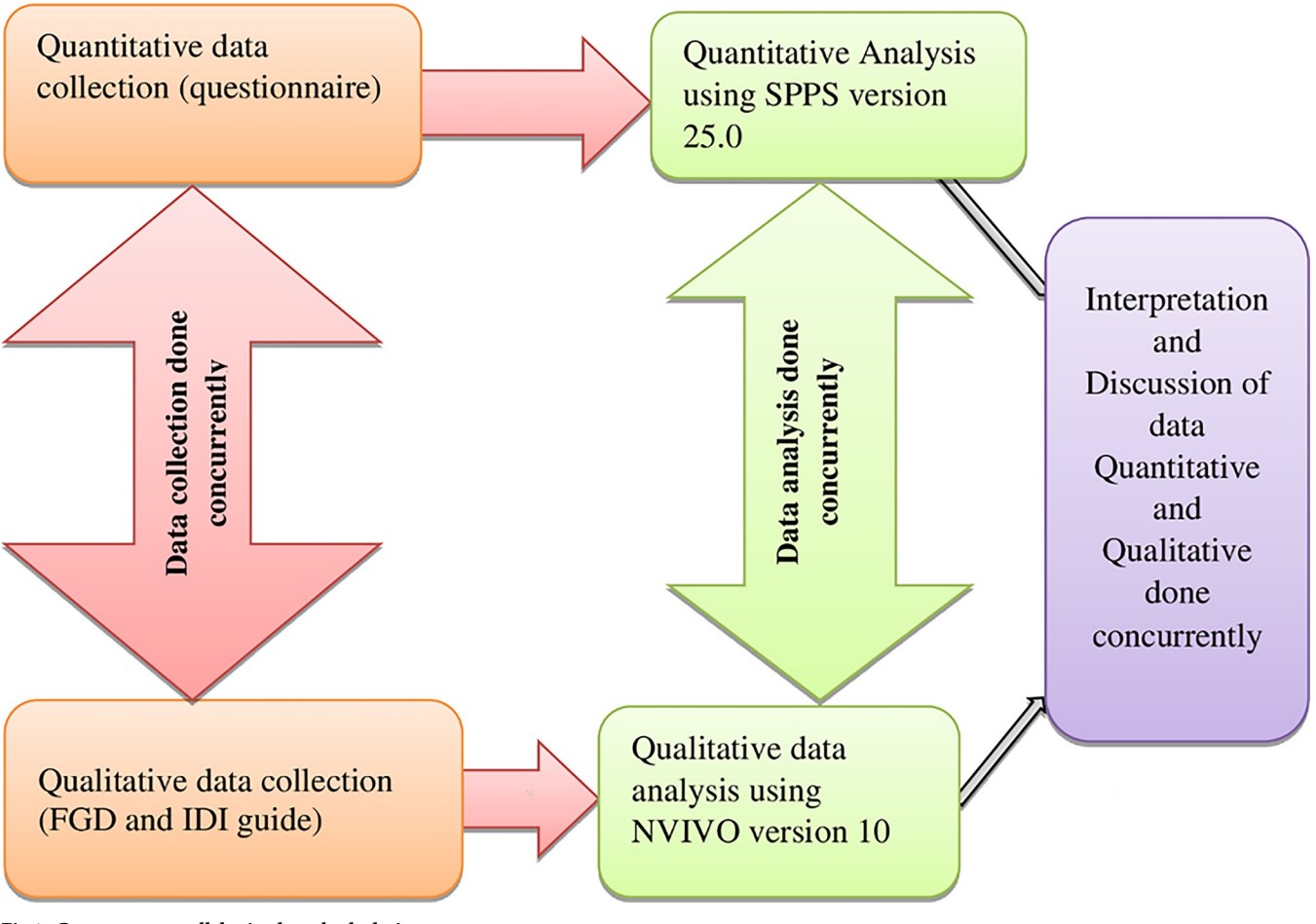

**Fig 1. Convergent parallel mixed methods design.**

county with similar morbidity patterns and a high likelihood of being influenced by the UHC rollout. This is because patients from the neighboring county may cross into the pilot county for medical treatment. The second county in each pair is geographically distant and does not share a border with any UHC pilot counties. This reduces the likelihood of patients seeking treatment in the pilot county, thereby creating a total of twelve counties for analysis.

### 2.3 Study population

Study population are Facility in-charges to assess human resources capacity and for qualitative arm are members of County Health Management Teams (CHMTs), Community Health Volunteers (CHVs); members of the health committees at the County Assembly, Community and implementing partners.

### 2.4 Sample size determination for quantitative arm

The Sample size calculations were done at County domain level by ensuring adequate representation of two facility ownership groups, and three KEPH facility levels. For qualitative arm, the study employed a specialized sample size formula. The sample size formula that allows for estimating the minimum sample size required to estimate a proportion with a pre-defined desired level of precision was employed [8]. This formula facilitates the determination of the

minimum sample size essential for estimating a proportion with a predefined level of precision. Adjustments were made to the calculated sample sizes to cater for design effect, non-response and correction for finite populations [Finite Population Correction Factor (FPC)]. A total of 791 health facilities were sampled. Required number of facilities per county was determined using square root allocation method. In each selected County, facilities were stratified by ownership and KEPH level. The required number of facilities per strata was determined using square root allocation method. Simple random sampling was applied to sample required number of facilities per strata.

Qualitative participants were purposively selected to balance between diversity, representativeness and expertise from various categories until saturation.

## 2.5 Sampling

**Quantitative arm:** Simple random sampling method was applied to sample required number of facilities per county.

**Qualitative components:** A total of 36 FGDs and 30 IDIs per county were conducted. Each FGD comprises of 8–10 participants.

## 2.6 Variables

**Outcomes:** The main outcome of the study is the availability and distribution of HRH in UHC pilot Counties and Non-UHC pilot Counties of Kenya.

**Exposures:** The main exposure of the study is the capacity of the health system to meet MOH staff norms and standard, in terms of cadres, levels and ownership.

**Predictors:** The availability and distribution of HRH across Counties, rural-urban and levels.

**Potential confounders**: Some potential confounders include the level of economic development, the prevalence of communicable and non-communicable diseases, the level of health literacy, and cultural factors that affect health-seeking behavior.

**Effect modifiers:** Some possible effect modifiers could include the level of decentralization of the health system, the type of health facility or provider, the level of community engagement, and the level of private sector participation in the health system. These factors may affect the capacity of the health system to provide HRH and implement UHC differently in different contexts or settings.

## 2.7 Data collection, management and analysis

Health facilities assessment tool adopted from the Kenya Service Availability and Readiness Assessment Mapping (SARAM) was administered to facility in-charge to assess the health facilities' capacity with respect to staffing. The tool was administered through an open-source software, KoBo ToolKit$^{TM}$ which is a digital data gathering platform. Data quality assurance was effected through testing of the collection tool in the pilot study. Also, the database design had inbuilt data quality checks that included appropriate skip patterns, logic and quality checks that prompted enumerators in real time for data entry errors. During data collection, the data manager routinely carried out data validation by profiling sample data for completeness and consistency. Any error identified was reported to the team lead for follow up and correction or verification on daily basis during the entire process. Data was uploaded into the cloud server and password protected, backed up regularly to avoid any loss or tampering and all study documentation and data files were access controlled for confidentiality with the principal investigator being the sole custodian of all data (hard and soft copies).

During quantitative data analysis, data cleaning and validation was done prior to statistical analysis using IBM SPSS version 27. To uncover the distribution structure of continuous study variables as well as identify outliers or unusually entered values, exploratory data analysis (EDA) technique was employed. For continuous variables descriptive statistics (means & standard deviations for normally distributed data and median and inter quartile range (IQR) for skewed data were computed. For categorical variables descriptive statistics in terms of proportions, frequency distributions and percentages were generated that constituted univariate analysis. Bivariate analysis was done using Pearson's Chi-square or Fisher Exact test based on the distribution of the mean expected count to compare HRM general service availability index between UHC pilot counties and Non-UHC pilot counties where significance threshold was established at $p < 0.05$. MS-Excel was also used to generate graphical presentations for various figures.

For the qualitative, data was collected using focus group discussions with Community Health Volunteers (CHVs); members of the health committees at the County Assembly, Community and implementing partners, In-depth interviews were conducted with members of county health management teams (CHMTs). Guides for Focus group discussions (FGDs) and In-depth interviews (IDIs) were used for data collection. It was moderated by the social scientists with qualitative data collection expertise assisted by local trained research assistants who audio record and take notes as backup. Its audio recordings were transcribed verbatim into MS-Word, data cleaning and preparation for analysis which was manually done based on content and thematic analytical framework. Familiarization allowed for the data to be organized and generation of themes, coding and interpretation of findings. Findings are described in summaries with supportive verbatim.

## 2.8 Validity and reliability

A pilot study and content validity test were done on the data collection to test its reliability and validity prior to the study data collection.

## 2.9 Ethical considerations

Scientific and ethical approval to conduct this study was obtained from the Kenya Medical Research Institute's Scientific and Ethical Review Unit **(KEMRI/SERU/CPHR/OO5/3945).** The Research permit was obtained from the National Council of Science Technology & Innovation (NASCOTI) and written authorization to conduct the research in the County was obtained from County Health Director and a letter of support from the Ministry of Health. Written informed consent were sought from participants prior to their participation in the study, done in a language that the participant can understand very well with the help of local research assistant with back-to-back translation. We took all necessary steps to ensure that participants' confidentiality was maintained throughout the study by assigning participants unique study identification numbers to ensure anonymity. All study documentation and materials were stored in locked file cabinets when not in use. Only investigators and project staffs have access to this information.

## 2.10 Response rate

Out of the 791 facilities randomly selected, a total of 746 Health facilities were assessed. Most of the health facilities that were not assessed were either not operational or were inaccessible during the study period. According to KEPH level distribution level 2, 3 and 4 are at 47.5 percent, 30.9 percent, 21.6 percent respectively. 75.0 percent of the selected facilities were public health facilities while 25.0 percent were non-public (private either for profit or not for profit).

**Table 1. Staffing norms across different health facility levels.**

| No. | Cadre | Level 5 | Level 4 | Level 3 | Level 2 | Level 1 |
|---|---|---|---|---|---|---|
| 1 | Generalist (non-specialist) medical doctors/ Medical Officers | 50 | 16 | 2 | 0 | 0 |
| 2 | Physician | 4 | 2 | 0 | 0 | 0 |
| 3 | Clinical Officer | 44 | 30 | 6 | 2 | 1 |
| 4 | Specialist medical doctors | | 1 | 0 | 0 | 0 |
| 5 | Psychiatrist | 4 | 2 | 0 | 0 | 0 |
| 6 | Any NCD specialist | 2 | 1 | 0 | 0 | 0 |
| 7 | Non-physician clinicians/paramedical professionals | 3 | 1 | 0 | 0 | 0 |
| 8 | Nutrition Professional | 12 | 8 | 4 | 2 | 1 |
| 9 | Pharmacy professional | 10 | 8 | 4 | 1 | - |
| 10 | Nursing professionals | 250 | 100 | 12 | 4 | 1 |
| 11 | Midwifery professionals | 60 | 20 | 6 | 0 | 0 |
| 12 | Laboratory professionals | 50 | 40 | 10 | 2 | 0 |
| 13 | Dental Specialist, Dentist and Community Oral Health Officer (COHO) | 30 | 15 | 7 | 2 | 1 |

## 2.11 Human resource for health norms and standards across different facility levels

Human resource for health norms and standards refers to the minimum and appropriate mix of human resources and infrastructure that is required to serve the expected populations at the different levels of the system with the defined health services. The study employed Service Availability and Readiness Assessment as a health facility assessment tool designed to assess and monitor the service availability and readiness of the health sector and to generate evidence to support the planning and managing of the health system [9]. The tool is designed as a systematic survey to generate a set of tracer indicators of service availability and readiness. HRM is one of the domains in the general service availability. Service availability refers to the physical presence of the delivery of services and encompasses health infrastructure, core health personnel and aspects of service utilization.

HRH general service availability index is generated on availability five tracer items that was Nurses, Clinician, Nutritionist, Laboratory technologist/officer and Pharmacist. The scores are 0–5, a score of five (*the health facility has all the five cadres represented/are available)* is used to generate the overall score of the HRH since the tracer indicators sampled during the survey were evenly distributed across all the KEPH levels of interest as defined by the ministry of health in the health workforce norms and standards to ensure that there is adequate and equitable distribution of human resources for health [7], as highlighted in Table 1 which includes clinical officers, nutritionists, pharmacists, nurses and laboratory professionals.

## 3. Results

The study assessed availability of different health professionals and a list of specific health workers across all levels. The HRH general service availability index was measured using five tracer items (Nurses, Clinician, Nutritionist, Laboratory technologist and Pharmacist) which is a minimum requirement across all levels of health facilities.

## 3.1 Health facilities that meet minimum staffing norms in UHC pilot counties and non-pilot counties

The survey established that the overall low HRH general service availability index at 17.2 percent, with non-public facilities (not owned by government) generally outperforming

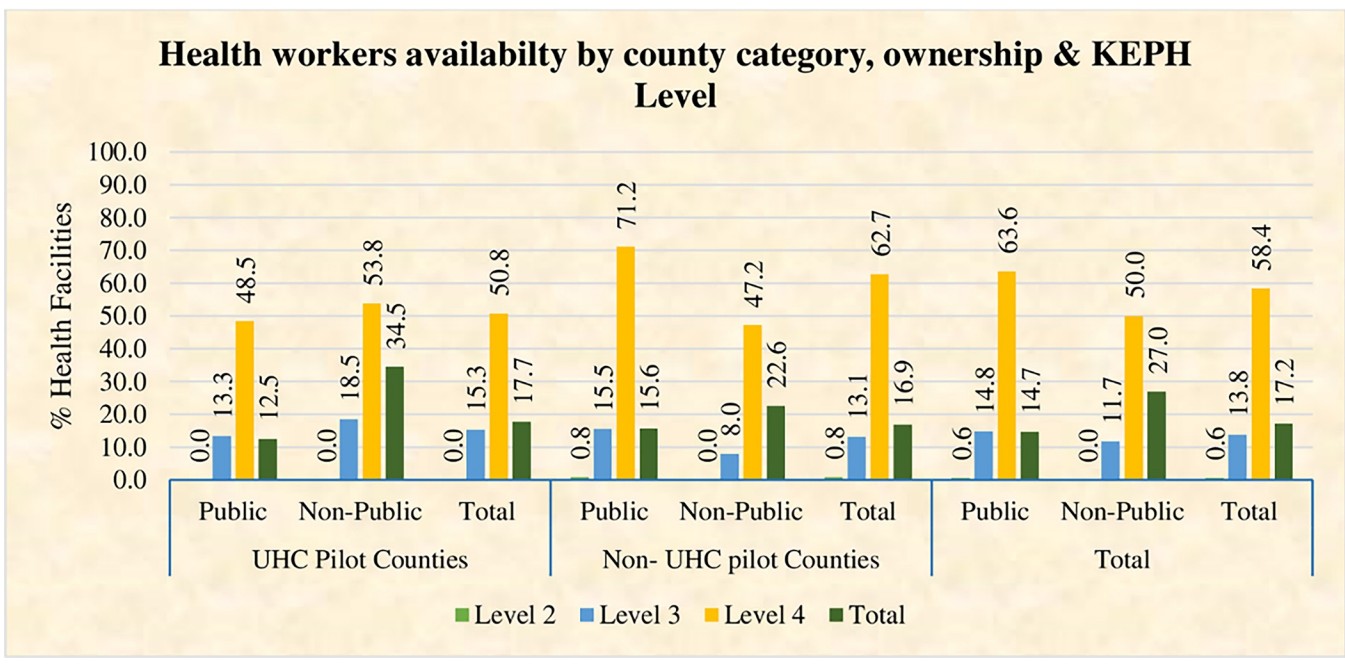

**Fig 2. Availability of health workers by ownership and KEPH Level in UHC pilot and non-UHC pilot counties.**

public facilities (owned by government) at 27.0 percent and 14.7 percent respectively as shown in Fig 1. The comparison between UHC pilot and non-UHC counties did not reveal any significant differences in HRM service availability, with UHC pilot Counties at 17.7% and Non-UHC pilot Counties at 16.9 percent with no statistical significance (P = 0.834). Additionally, a relatively small percentage of health facilities 17.2 percent met the minimum staff norms for health workers, indicating a shortage of personnel in most facilities as shown in Fig 2.

### 3.2 Health facilities that meet minimum staffing norms as per the location of the health facilities

In rural areas, 11.1 percent of facilities meet staffing norms, whereas facilities in urban and peri-urban areas stand at 31.8% and 30.4%, as indicated in Fig 3.

### 3.3 Health worker availability index by county, ownership and KEPH Level

The availability of the health workers varied with various dimensions like specific County, ownership and level. In respect to Counties, only three counties had more than 20.0 percent, that is Kisumu 21.2 percent, Bungoma 23.3 percent and Kitui 21.0 percent. The Counties with lowest percentage of meeting staff norms are Bomet 9.7 percent and Isiolo 10.3 percent. Further, with respect to public health facilities, only one county that had more than 20.0 percent in the availability of health workers was Kitui County at 21.2 percent with the bottom Isiolo County 4.5 percent. The availability of health workers increased with advancement in the KEPH Levels where the Level 2 (primary facilities) across all counties except Taita Taveta 8.3 percent, didn't meet MOH staffing norms (0.0 percent) as shown in Table 2.

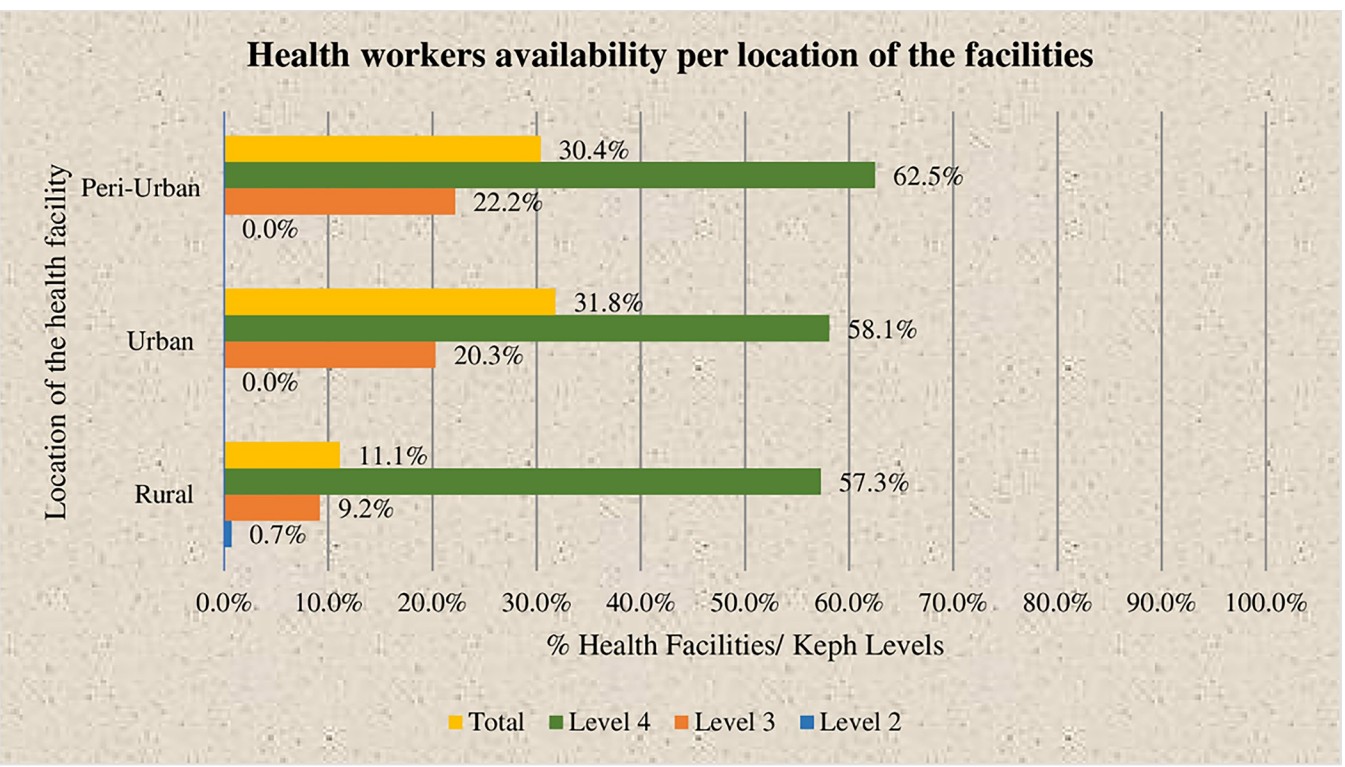

**Fig 3. Availability of health workers meeting staff norms as per location of the facilities.**

### 3.4 Availability of health workers' specific cadres by government owned and non-governmental

The findings reveal the variation in the availability of health workers across specific cadres. The nursing profession demonstrates a high compliance rate at 93.3 percent, while the clinical officer and laboratory professions exhibit moderate compliance at 52.3 percent and 55.2 percent respectively. However, there are notable shortages in the nutrition and pharmacy professions at 21.6 percent and Pharmacy 33.0 percent as profiled in Table 3.

### 3.5 Availability of health worker specific cadres by County

The findings demonstrate the variation in availability scores by cadre across counties, highlighting discrepancies in compliance with staff norms. Kisumu County stands out as the only county where all assessed public health facilities met the staff norm for the nursing profession as shown in Table 4.

### 3.6 Qualitative findings

**3.6.1 Investments in health through hiring HRH cadres.** Several County health heads expressed that they had initiated plans to augment their healthcare workforce across all levels in order to prepare for the implementation of Universal Health Coverage as shown in below verbatim.

*". . .a key priority for this year is recruitment of health personnel. Where want to recruit 10 medical officers, 5 specialists another 2 also that we want to recruit. We have put in priority the recruitment of more staffs" [Kisumu, IDI FIC]*

**Table 2. Health worker availability index by county, ownership and KEPH level.**

| County category | County | Managing Authority | Total (n = 746) | KEPH Level (%) | | | |
|---|---|---|---|---|---|---|---|
| | | | | Level 2 | Level 3 | Level 4 | Total |
| *UHC Pilot Counties* | Isiolo | Public | 22 | 0.0 | 0.0 | 50.0 | **4.5** |
| | | Non-Public | 7 | 0.0 | 20.0 | 100.0 | **28.6** |
| | | Total | 29 | 0.0 | 11.1 | 66.7 | **10.3** |
| | Kisumu | Public | 46 | 0.0 | 0.0 | 30.4 | **15.2** |
| | | Non-Public | 20 | 0.0 | 16.7 | 42.9 | **35.0** |
| | | Total | 66 | 0.0 | 7.1 | 35.1 | **21.2** |
| | Machakos | Public | 53 | 0.0 | 26.7 | 100.0 | **15.1** |
| | | Non-Public | 18 | 0.0 | 27.3 | 50.0 | **33.3** |
| | | Total | 71 | 0.0 | 26.9 | 70.0 | **19.7** |
| | Nyeri | Public | 55 | 0.0 | 11.1 | 100.0 | **10.9** |
| | | Non-Public | 10 | 0.0 | 0.0 | 80.0 | **40.0** |
| | | Total | 65 | 0.0 | 8.7 | 88.9 | **15.4** |
| *Non-UHC Pilot Counties* | Bungoma | Public | 42 | 0.0 | 10.0 | 70.0 | **19.0** |
| | | Non-Public | 18 | 0.0 | 10.0 | 62.5 | **33.3** |
| | | Total | 60 | 0.0 | 10.0 | 66.7 | **23.3** |
| | Homa Bay | Public | 60 | 0.0 | 0.0 | 60.0 | **15.0** |
| | | Non-Public | 18 | 0.0 | 10.0 | 57.1 | **27.8** |
| | | Total | 78 | 0.0 | 3.4 | 59.1 | **17.9** |
| | Kitui | Public | 66 | 0.0 | 13.3 | 92.3 | **21.2** |
| | | Non-Public | 15 | 0.0 | 0.0 | 75.0 | **20.0** |
| | | Total | 81 | 0.0 | 8.0 | 88.2 | **21.0** |
| | Meru | Public | 51 | 0.0 | 8.3 | 61.5 | **17.6** |
| | | Non-Public | 21 | 0.0 | 18.2 | 33.3 | **23.8** |
| | | Total | 72 | 0.0 | 13.0 | 50.0 | **19.4** |
| | Nyandarua | Public | 51 | 0.0 | 23.8 | 100.0 | **13.7** |
| | | Non-Public | 7 | 0.0 | 0.0 | 100.0 | **14.3** |
| | | Total | 58 | 0.0 | 20.8 | 100.0 | **13.8** |
| | Taita Taveta | Public | 41 | **8.3** | 8.3 | 80.0 | **17.1** |
| | | Non-Public | 6 | 0.0 | 100.0 | 100.0 | **100.0** |
| | | Total | 47 | 8.3 | 7.1 | 44.4 | **14.9** |
| | West Pokot | Public | 54 | 0.0 | 37.5 | 100.0 | **13.0** |
| | | Non-Public | 3 | 0.0 | 100.0 | 100.0 | **100.0** |
| | | Total | 57 | 0.0 | 30.0 | 80.0 | **12.3** |
| | Bomet | Public | 57 | 0.0 | 30.8 | 25.0 | **8.8** |
| | | Non-Public | 5 | 0.0 | 0.0 | 50.0 | **20.0** |
| | | Total | 62 | 0.0 | 26.7 | 33.3 | **9.7** |

*"Prioritized the training of healthcare workers where in every ward a good number of health workers that are going around. They are teaching members of the public on the needs of primary health care"* [Bomet, IDI FIC]

**3.6.2 Shortage of human resource for health but high demand for services.** Multiple participants have reported a deficit of healthcare workers in relation to the increased demand for healthcare services, as stated below.

**Table 3. Specific cadres' availability by ownership and KEPH level.**

| Managing Authority | Cadres | KEPH Level (%) | | | |
|---|---|---|---|---|---|
| | | Level 2 | Level 3 | Level 4 | Total |
| **Public (Government owned)** | Clinical Officer | 12.8 | 78.7 | 100.0 | **44.3** |
| | Nutrition Professional | 1.7 | 24.5 | 72.7 | **19.4** |
| | Pharmacy professional | 4.1 | 34.2 | 87.9 | **25.8** |
| | Nursing professional | 92.4 | 96.1 | 100.0 | **94.6** |
| | Laboratory professional | 21.2 | 72.9 | 96.0 | **47.0** |
| **Non-Public (Not owned by Government)** | Clinical Officer | 55.6 | 81.8 | 91.9 | **84.5** |
| | Nutrition Professional | 0.0 | 16.9 | 51.6 | **30.4** |
| | Pharmacy professional | 11.1 | 45.5 | 90.3 | **62.2** |
| | Nursing professional | 77.8 | 84.4 | 91.9 | **87.2** |
| | Laboratory professional | 44.4 | 85.7 | 98.4 | **88.5** |
| **Total** | Clinical Officer | 13.9 | 79.7 | 96.9 | **52.3** |
| | Nutrition Professional | 1.7 | 22.0 | 64.6 | **21.6** |
| | Pharmacy professional | 4.2 | 37.9 | 88.8 | **33.0** |
| | Nursing professional | 92.1 | 92.2 | 96.9 | **93.2** |
| | Laboratory professional | 21.8 | 77.2 | 96.9 | **55.2** |

*"The health care workers here are few. The government can help us by employing more health care workers so that they can help each other when there are so many patients. And if there is a mother at home, the doctor can go there and attend to them"* [FGD, Elderly, West Pokot]

*"The problem that we have as he has mentioned is shortage of nurses. The second one is that the facility has water shortage and also drugs shortages"* [FGD, Men, Isiolo]

*"Previously, the government employed new staff and it will relieve us in a way because staff shortage is overwhelming. Like a clinician you see over 100 patients in a day but at least when they employ more staff you won't be exhausted at the end of the day and you also offer quality services"* [IDI, FIC Meru]

*"One of the challenges is that there are no treatment services on Sunday. In most cases, there is a shortage of doctors who come to serve on that day. Sometimes it is not good because this facility has been upgraded and we expected they would add more doctors. . .."* [FGD, Men, Nyandarua]

**3.6.3 Efforts of hiring of HRH carders but no systems of managing resource.** The analysis of the discussion with county assembly members revealed that a significant majority of them recognized the dearth of healthcare professionals. Moreover, a subset of them expressed intentions to give priority to the recruitment of additional healthcare workers across all professional categories as part of their preparation for the implementation of Universal Health Coverage.

*"On human resource, definitely to acquire more additional staff. More in terms of recruitment and all that. then prevention generally. There was a quest to invest more on prevention so that we try as much as possible to reduce now the workload in our facilities"* (FGD, MCA, Meru)

*". . .A key priority for this year is recruitment of health personnel where want to recruit 10 medical officers, 5 specialists, and another 2. We have put in priority the recruitment of more staffs"* (FGD, MCA, Kisumu)

**Table 4. Specific cadres' availability by county and KEPH level.**

| County Category | Tracer Elements (Cadres) | KEPH Level (%) | | | |
|---|---|---|---|---|---|
| | | *Level 2* | *Level 3* | *Level 4* | *Total* |
| **Bungoma** | Clinical Officer | 4.5 | 85.0 | 94.4 | **58.3** |
| | Nutrition Professional | 0.0 | 10.0 | 66.7 | **23.3** |
| | Pharmacy professional | 0.0 | 40.0 | 77.8 | **36.7** |
| | Nursing professional | 86.4 | 85.0 | 88.9 | **86.7** |
| | Laboratory professional | 9.1 | 90.0 | 94.4 | **61.7** |
| **Homa Bay** | Clinical Officer | 18.5 | 69.0 | 100.0 | **60.3** |
| | Nutrition Professional | 0.0 | 13.8 | 86.4 | **29.5** |
| | Pharmacy professional | 0.0 | 20.7 | 72.7 | **28.2** |
| | Nursing professional | 96.3 | 93.1 | 95.5 | **94.9** |
| | Laboratory professional | 3.7 | 55.2 | 81.8 | **44.9** |
| **Isiolo** | Clinical Officer | 17.6 | 66.7 | 100.0 | **41.4** |
| | Nutrition Professional | 0.0 | 11.1 | 100.0 | **13.8** |
| | Pharmacy professional | 17.6 | 44.4 | 66.7 | **31.0** |
| | Nursing professional | 76.5 | 66.7 | 100.0 | **75.9** |
| | Laboratory professional | 29.4 | 77.8 | 100.0 | **51.7** |
| **Kisumu** | Clinical Officer | 13.3 | 78.6 | 97.3 | **74.2** |
| | Nutrition Professional | 0.0 | 14.3 | 37.8 | **24.2** |
| | Pharmacy professional | 13.3 | 35.7 | 91.9 | **62.1** |
| | Nursing professional | 100.0 | 100.0 | 100.0 | **100.0** |
| | Laboratory professional | 13.3 | 57.1 | 100.0 | **71.2** |
| **Kitui** | Clinical Officer | 7.7 | 84.0 | 100.0 | **50.6** |
| | Nutrition Professional | 0.0 | 16.0 | 88.2 | **23.5** |
| | Pharmacy professional | 2.6 | 20.0 | 100.0 | **28.4** |
| | Nursing professional | 89.7 | 96.0 | 100.0 | **93.8** |
| | Laboratory professional | 5.1 | 76.0 | 100.0 | **46.9** |
| **Machakos** | Clinical Officer | 8.6 | 92.3 | 90.0 | **50.7** |
| | Nutrition Professional | 0.0 | 38.5 | 70.0 | **23.9** |
| | Pharmacy professional | 0.0 | 53.8 | 100.0 | **33.8** |
| | Nursing professional | 100.0 | 96.2 | 100.0 | **98.6** |
| | Laboratory professional | 17.1 | 96.2 | 100.0 | **57.7** |
| **Meru** | Clinical Officer | 18.5 | 73.9 | 95.5 | **59.7** |
| | Nutrition Professional | 0.0 | 13.0 | 54.5 | **20.8** |
| | Pharmacy professional | 11.1 | 43.5 | 95.5 | **47.2** |
| | Nursing professional | 92.6 | 91.3 | 95.5 | **93.1** |
| | Laboratory professional | 37.0 | 82.6 | 100.0 | **70.8** |
| **Nyandarua** | Clinical Officer | 6.5 | 75.0 | 100.0 | **39.7** |
| | Nutrition Professional | 3.2 | 29.2 | 100.0 | **19.0** |
| | Pharmacy professional | 0.0 | 37.5 | 100.0 | **20.7** |
| | Nursing professional | 96.8 | 100.0 | 100.0 | **98.3** |
| | Laboratory professional | 35.5 | 83.3 | 100.0 | **58.6** |
| **Nyeri** | Clinical Officer | 18.2 | 95.7 | 88.9 | **55.4** |
| | Nutrition Professional | 0.0 | 13.0 | 88.9 | **16.9** |
| | Pharmacy professional | 3.0 | 39.1 | 100.0 | **29.2** |
| | Nursing professional | 93.9 | 91.3 | 100.0 | **93.8** |
| | Laboratory professional | 21.2 | 78.3 | 100.0 | **52.3** |

*(Continued)*

**Table 4.** (Continued)

| County Category | Tracer Elements (Cadres) | KEPH Level (%) | | | |
|---|---|---|---|---|---|
| | | Level 2 | Level 3 | Level 4 | Total |
| Taita Taveta | Clinical Officer | 20.8 | 78.6 | 100.0 | **53.2** |
| | Nutrition Professional | 12.5 | 14.3 | 44.4 | **19.1** |
| | Pharmacy professional | 16.7 | 50.0 | 88.9 | **40.4** |
| | Nursing professional | 100.0 | 85.7 | 88.9 | **93.6** |
| | Laboratory professional | 66.7 | 92.9 | 100.0 | **80.9** |
| West Pokot | Clinical Officer | 16.7 | 60.0 | 100.0 | **31.6** |
| | Nutrition Professional | 2.4 | 60.0 | 80.0 | **19.3** |
| | Pharmacy professional | 0.0 | 50.0 | 100.0 | **17.5** |
| | Nursing professional | 92.9 | 80.0 | 100.0 | **91.2** |
| | Laboratory professional | 21.4 | 60.0 | 100.0 | **35.1** |
| Bomet | Clinical Officer | 17.1 | 80.0 | 100.0 | **40.3** |
| | Nutrition Professional | 2.4 | 46.7 | 50.0 | **17.7** |
| | Pharmacy professional | 2.4 | 40.0 | 66.7 | **17.7** |
| | Nursing professional | 80.5 | 100.0 | 100.0 | **87.1** |
| | Laboratory professional | 14.6 | 66.7 | 100.0 | **35.5** |

The facility in-charges also acknowledged the shortage of healthcare workers as a hindrance to the successful implementation of Universal Health Coverage (UHC). They highlighted that this scarcity directly impacts hospital waiting times, the quality of care provided, and even influences health-seeking behaviors, as indicated in the following quote.

*"Number one is workload. You have seen here. As a clinician I am supposed to see maybe 10 patients a day. You have seen from morning to evening am going towards 70 one person. So, number one is human resource. . ." (IDI, FIC, Bungoma)*

*"The barriers that will limit UHC services or whatever is staffing. If they cannot address the issue of staffing, there is no way it will succeed. . ." (IDI, FIC, Bungoma)*

*"Definitely. One the services will be over stretched, yeah. . .Talk of the health care providers I don't think they will be enough to handle the numbers. The service delivery will be so much affected if we are not very well prepared" (IDI, IP, Homabay)*

*"Yes. In fact, we are very grateful for the staffing example since the corona pandemic (COVID-19), we have been given 3 staffs. They were given courtesy of UHC and funded by the program" (IDI, FIC, Taita Taveta)*

## 4. Discussion

The capacity of human resources for health is a critical factor in the implementation of Universal Health Coverage in Kenya. The HRH availability index of 17.2 percent underscores a considerable shortfall in health workers' availability across all surveyed health facilities. This finding raises concerns about the adequacy of the healthcare workforce and its potential impact on the quality and accessibility of healthcare services. Another study conducted by the Ministry of Health in Kenya in 2015 also found similar results. The study revealed that there were only 7,931 doctors, 16,241 nurses, and 1,327 clinical officers working in the country, this results in a ratio of 1.6 healthcare workers per 1,000 people, significantly falling below the

WHO-recommended minimum threshold of 4.45 health workers (including physicians, nurses, and midwives) per 1,000 people [10]. The findings highlight the need for comprehensive interventions to address the shortage of health care professional and compliance with staff norms for health workers. It is crucial to prioritize investments in the healthcare workforce, especially in the public sector, and ensure equitable distribution of health workers across regions and facility types. By doing so, the healthcare system can better meet the population's healthcare needs, enhance service quality, and ultimately improve health outcomes.

The UHC pilot counties and non-UHC counties showed comparable HRH availability indexes (Fig 3). This finding suggests that the implementation of the UHC pilot program did not significantly impact the availability of health workers. This is consistent with other research in this area. For example, a study conducted by the World Bank in 2019 found that there was no significant difference in health outcomes between UHC pilot and non-pilot counties in Kenya [11]. Similarly, a study conducted by the Institute for Health Metrics and Evaluation (IHME) in 2020 found that the introduction of UHC in Kenya did not have a significant impact on overall health outcomes [12]. It is important to further investigate the factors that may have influenced this lack of difference and identify opportunities to strengthen the workforce within the UHC framework to enhance the effectiveness.

The findings underscore the significant staffing challenges faced by health facilities in rural areas. It reveals significant staffing disparities among health facilities based on their location. The lower compliance rates in rural areas indicate a pressing need to address the staffing challenges faced by rural health facilities. Rural have more shortage of worker than urban and peri-urban, a similar study by ministry of health shows the shortage of healthcare workers is particularly acute in rural areas, where only 23% of the population has access to healthcare services [10]. Several factors contribute to the lower staffing levels in rural areas as cited in qualitative arm of the study. Firstly, rural regions often struggle with attracting and retaining healthcare professionals. Additionally, the distribution of healthcare resources tends to be concentrated in urban areas, making it more challenging to allocate an adequate number of health workers to rural settings. Addressing the staffing shortages in rural areas requires a multifaceted approach. Efforts should be made to enhance recruitment strategies that provide incentive to the healthcare professionals interested in serving in rural communities. This may include providing incentives and professional development opportunities.

Public facilities have lower availability of health workers than private facilities. This discrepancy suggests that public health facilities face greater challenges in meeting staffing requirements. Several studies have reported similar findings to the observation that public health facilities have lower availability of health workers than private facilities. For instance, a study found that private health facilities had higher numbers of health workers per facility compared to public facilities in Ethiopia [13]. Similarly, a study in Kenya found that private health facilities had more health workers and better staff distribution than public facilities. It emphasizes the need for interventions to address this disparity and ensure adequate staffing levels in public healthcare facilities, which serve a significant portion of the population, particularly those with limited financial resources [14].

Primary health care facilities (Level 2) have lower availability of health worker compared to Secondary health care facilities (Level 3 and 4). A large number of level 2 and 3 both public and Non-public facilities did not meet the minimum HRH norms. Most level 4 hospitals, whether public or private health facilities, had the required staff. Similarly, a study found that there were fewer health workers per population in primary health care facilities in Kenya compared to secondary health care facilities [15]. The finding underscores the need for tailored interventions to address this disparity. By improving resource allocation, incentivizing primary care practice, and investing in the education and training of primary care professionals,

it is possible to enhance the availability of health workers in primary care settings, strengthen the primary healthcare system, and ultimately improve health outcomes for the population.

The finding that certain cadres of health workers are more available than others, with nurses being the most available and nutritionists being the least available, highlights a significant disparity in the distribution and availability of different healthcare professionals. The finding is consistent with previous research. A study conducted [16] in Kenya found that nurses were the most widely available health workers, followed by clinical officers and medical officers. Similarly, a study [17] found that nurses were the most widely available health workers in many low- and middle-income countries. It is essential for policymakers and healthcare authorities to recognize the variations in the availability of health workers' cadres and prioritize investments and interventions accordingly. Strategies to address these shortages and improve the availability of health workers in all cadres are essential for a comprehensive and well-functioning healthcare system.

This finding underscores the need to address county-specific challenges and implementation of tailored strategies to strengthen specific cadres in areas where compliance with staff norms is low. Sharing best practices and learning from successful counties can contribute to improving the availability of health workers with the required skills set and enhancing healthcare delivery across all regions. By addressing these disparities and ensuring that health workers possess the necessary skills, the quality and effectiveness of healthcare services can be improved, leading to better health outcomes for the population.

## 5. Conclusions

This finding underscores the existence of a substantial gap in the availability of healthcare professionals as a critical challenge that must be addressed to achieve the goal of Universal Health Coverage in Kenya. Efforts should be made to bridge this gap and enhance the overall healthcare system.

The comparison between UHC pilot counties and non-pilot UHC counties revealed similar health workforce availability indexes. It highlights the need to explore additional interventions or strategies to address the existing gaps to ensure meeting staff norms.

The shortage of healthcare workers is more pronounced in rural regions compared to urban and peri-urban areas. Implementing strategies to attract and retain healthcare professionals in rural areas is crucial to ensure equitable access to quality healthcare for all populations.

The low availability of health workers in public facilities may impact the accessibility and quality of healthcare services provided to the general population. The underlying factors contributing to this disparity are multifaceted and require comprehensive interventions at policy, financial, and organizational levels.

Addressing the shortage of health workers in primary healthcare settings (level 2) is crucial to ensure adequate and accessible healthcare services at the community level. Efforts should focus on recruiting and retaining healthcare professionals in primary healthcare facilities to meet the healthcare needs of the population.

## 6. Study limitation

The study employed a cross-sectional design, capturing data at a specific point in time. This limits the ability to establish causal relationships or assess changes in HRH availability over time. The data on the availability of human resources for health was collected using a customized tool and relied on self-reporting by the health facilities. There are no details of the actual human resources adequacy, educational background or training experiences assessed. It misses

a budget and futuristic financing landscape analysis to determine the feasibility of meeting the staffing norms. However, mixed-methods approaches and enhanced robustness and reliability of its findings, the study nuanced understanding of HRH availability.

## Acknowledgments

The authors extend their sincere gratitude to the Governors, members of the County Executive Committee (CEC)–Health, the Director of Medical Services, and the entire staff of the Ministry of Health for their support and cooperation throughout the research. We also express their appreciation to the implementing partners, private and faith-based healthcare services providers for their valuable insights and assistance. We are thankful to all the participants in the survey and everyone who contributed directly or indirectly to the success of the study, acknowledging their invaluable support and cooperation.

## Author Contributions

**Conceptualization:** Ismail Adow Ahmed, David Mathu, Antony Macharia, Judy Mwai, Priscah Otambo, Violet Wanjihia, Joseph Mutai, Sharon Mokua, Lilian Nyandieka, Elizabeth Echoka, Doris Njomo, Zipporah Bukania.

**Data curation:** Ismail Adow Ahmed, James Kariuki, David Mathu, Stephen Onteri, Antony Macharia, Joseph Mutai, Sharon Mokua, Zipporah Bukania.

**Formal analysis:** Ismail Adow Ahmed, James Kariuki, Antony Macharia, Judy Mwai, Priscah Otambo, Joseph Mutai, Lilian Nyandieka, Elizabeth Echoka, Zipporah Bukania.

**Funding acquisition:** Zipporah Bukania.

**Investigation:** Ismail Adow Ahmed, Stephen Onteri, Judy Mwai, Zipporah Bukania.

**Methodology:** Ismail Adow Ahmed, James Kariuki, David Mathu, Stephen Onteri, Antony Macharia, Priscah Otambo, Joseph Mutai, Zipporah Bukania.

**Project administration:** Ismail Adow Ahmed, Zipporah Bukania.

**Resources:** Zipporah Bukania.

**Software:** Zipporah Bukania.

**Supervision:** Zipporah Bukania.

**Validation:** Ismail Adow Ahmed, James Kariuki, Stephen Onteri, Antony Macharia, Judy Mwai, Priscah Otambo, Joseph Mutai, Zipporah Bukania.

**Visualization:** Ismail Adow Ahmed, James Kariuki, Antony Macharia, Judy Mwai, Priscah Otambo, Joseph Mutai, Zipporah Bukania.

**Writing – original draft:** Ismail Adow Ahmed, James Kariuki, David Mathu, Stephen Onteri, Antony Macharia, Judy Mwai, Priscah Otambo, Violet Wanjihia, Joseph Mutai, Doris Njomo, Zipporah Bukania.

**Writing – review & editing:** Ismail Adow Ahmed, James Kariuki, David Mathu, Stephen Onteri, Antony Macharia, Judy Mwai, Priscah Otambo, Violet Wanjihia, Joseph Mutai, Sharon Mokua, Lilian Nyandieka, Elizabeth Echoka, Doris Njomo, Zipporah Bukania.

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
