## [Decision Letter · Decision Letter 0]

23 May 2023

PONE-D-23-11190Health systems’ capacity in availability of human resource for health towards implementation of Universal Health Coverage in KenyaPLOS ONE

Dear Dr. Ahmed,

Thank you for submitting your manuscript to PLOS ONE. After careful consideration, we feel that it has merit but does not fully meet PLOS ONE’s publication criteria as it currently stands. Therefore, we invite you to submit a revised version of the manuscript that addresses the points raised during the review process.

We look forward to receiving your revised manuscript.

Kind regards,

Philipos Petros Gile, MA

Academic Editor

PLOS ONE

Journal Requirements:

4. Please include a caption for figure 1.

Reviewers' comments:

Reviewer's Responses to Questions

**Comments to the Author**

1. Is the manuscript technically sound, and do the data support the conclusions?

Reviewer #1: Partly

Reviewer #2: Partly

2. Has the statistical analysis been performed appropriately and rigorously? 

Reviewer #1: Yes

Reviewer #2: Yes

3. Have the authors made all data underlying the findings in their manuscript fully available?

Reviewer #1: Yes

Reviewer #2: No

4. Is the manuscript presented in an intelligible fashion and written in standard English?

Reviewer #1: Yes

Reviewer #2: Yes

5. Review Comments to the Author

Reviewer #1: The article titled "Health systems’ capacity in availability of human resource for health towards implementation of Universal Health Coverage in Kenya" presents important findings on the availability of human resources for health (HRH) in Kenya and its potential impact on the implementation of universal health coverage (UHC). However, there are several areas of concern that need to be addressed.

Firstly, the study design and methodology should have been more detailed and transparent, including the sampling strategy and the criteria used to select the study sites. Additionally, the article lacks a clear description of the statistical methods used for data analysis and does not provide any information on the reliability and validity of the data collection tools.

Secondly, the article fails to provide a comprehensive analysis of the factors influencing the availability of HRH in Kenya. The article highlights the shortage and maldistribution of HRH in the country, but it does not explore the underlying causes of these challenges.

Thirdly, the article lacks a clear discussion on the potential implications of the findings for the implementation of UHC in Kenya. Although the article acknowledges the importance of HRH for the delivery and uptake of health services, it does not provide a clear link between the availability of HRH and the achievement of UHC.

Finally, the article could have provided a more critical reflection on the limitations and implications of the study. For instance, the article could have discussed the potential biases and limitations of the data collection tools, the possible impact of the COVID-19 pandemic on the availability of HRH, and the implications of the findings for other low- and middle-income countries facing similar challenges.

Reviewer #2: Thanks for sharing the manuscript. The paper provide part of a larger survey that focus on study Health systems’ capacity in availability of human resource for health towards implementation of Universal Health Coverage in Kenya. I hope the following suggestions be helpful:

1. The general format of paper need to revise and adjust based on essentials of Journal.

2. Abstract: The abstract needs a purposeful revisin in terms of both; form and content

3. Add appropriate key words based on MeSH categories of PubMed

4. Introduction: Introduction need to revision that mainly focus on the limitations/ gaps of evidence and justification of present study.

5. Methods: Refer to main characteristics of original survey and cite to it’s protocol study and related extracted papers.

6. Results: The visualization of results need to more attempts for better attraction

7. Results: Try to present analytical results and not limit the report to the description of indicators

8. Discussion: Complete the discussion by comparing the findings with those of other studies and applying an analysis of the causes of the differences in results

9. Discussion: Point to many solutions to delimit the challenges ahead as practical implication of findings.

10. Conclusion should be based on the net findings and proposing some general suggestions would not be practical.

6. PLOS authors have the option to publish the peer review history of their article (what does this mean?). If published, this will include your full peer review and any attached files.

Reviewer #1: No

Reviewer #2: No

---

## [Author Response · Author response to Decision Letter 0]

28 Jul 2023

I have responded to the reviewer is response to reviewers letter attached.

---

## [Decision Letter · Decision Letter 1]

12 Oct 2023

PONE-D-23-11190R1Health systems’ capacity in availability of human resource for health towards implementation of Universal Health Coverage in KenyaPLOS ONE

Dear Dr. Ahmed,

Thank you for submitting your manuscript to PLOS ONE. After careful consideration, we feel that it has merit but does not fully meet PLOS ONE’s publication criteria as it currently stands. Therefore, we invite you to submit a revised version of the manuscript that addresses the points raised during the review process.

Please submit your revised manuscript with point-by-point response to the reviewer by Nov 26 2023 11:59PM. If you will need more time than this to complete your revisions, please reply to this message or contact the journal office at plosone@plos.org. Please include the following items when submitting your revised manuscript:A rebuttal letter that responds to each point raised by the academic editor and reviewer(s). You should upload this letter as a separate file labeled 'Response to Reviewers'.A marked-up copy of your manuscript that highlights changes made to the original version. You should upload this as a separate file labeled 'Revised Manuscript with Track Changes'.An unmarked version of your revised paper without tracked changes. You should upload this as a separate file labeled 'Manuscript'.

We look forward to receiving your revised manuscript.

Kind regards,

Philipos Petros Gile, MA

Academic Editor

PLOS ONE

Reviewers' comments:

Reviewer's Responses to Questions

**Comments to the Author**

1. If the authors have adequately addressed your comments raised in a previous round of review and you feel that this manuscript is now acceptable for publication, you may indicate that here to bypass the “Comments to the Author” section, enter your conflict of interest statement in the “Confidential to Editor” section, and submit your "Accept" recommendation.

Reviewer #3: (No Response)

Reviewer #4: (No Response)

Reviewer #5: (No Response)

2. Is the manuscript technically sound, and do the data support the conclusions?

Reviewer #3: Partly

Reviewer #4: No

Reviewer #5: Partly

3. Has the statistical analysis been performed appropriately and rigorously? 

Reviewer #3: I Don't Know

Reviewer #4: No

Reviewer #5: Yes

4. Have the authors made all data underlying the findings in their manuscript fully available?

The PLOS Data policy requires authors to make all data underlying the findings described in their manuscript fully available without restriction, with rare exception (please refer to the Data Availability Statement in the manuscript PDF file). The data should be provided as part of the manuscript or its supporting information, or deposited to a public repository. For example, in addition to summary statistics, the data points behind means, medians and variance measures should be available. If there are restrictions on publicly sharing data—e.g. participant privacy or use of data from a third party—those must be specified

Reviewer #3: Yes

Reviewer #4: Yes

Reviewer #5: Yes

5. Is the manuscript presented in an intelligible fashion and written in standard English?

Reviewer #3: No

Reviewer #4: No

Reviewer #5: Yes

6. Review Comments to the Author

Reviewer #3: The topic of study is relevant, timely, and interesting. The authors have done a lot of work but that is not adequately reflected in the manuscript. There are queries that need to be clarified. The manuscript needs revision.

Define all concepts, preferably quoting government documents, and place them in context.

Expand all abbreviations - Use the expanded version at the first instance, immediately followed by the abbreviation in brackets; and then start using the abbreviation thereafter. This included even UHC! Do not leave readers to guess!

Expand abbreviations KEPH, HRM!

'Methodology'

Define concepts - KEPH Levels 2, 3, 4, etc. What do they mean, how do they differ from each other? Quote from Ministry of Health (MoH) documents.

Define UHC pilot counties and non-UHC pilot countries; quote from MoH document. Explain the extent of funding provided to facilities.

What is meant by 'targeted'? e.g., Page 5, Line 131; and elsewhere.

Page 6; Line 791 (sampling) - How did the authors arrive at this number? Procedure and criteria?

Why simple random sampling, and not stratified random sampling - if KEPH level distribution of the facilities are available? How was proportional representation ensured?

Lines 223-4: If this tool is a published document, give reference and give bibliographic information under 'References'.

In what language was the questionnaire (and informed consent) administered to various categories of respondents? Was the questionnaire translated into local language (and back-translated)?

Page 5, Line 119 - Have a closer look at this heading: study population. Where are you describing your 'respondents', how many from each category; and the method by which you arrived at these numbers?

What is meant by county in Kenyan parlance? Equivalent to a district, OR State / Province?

What is the total number of counties? How many were selected? How? Criteria? Consider drawing a single diagram showing the entire sampling procedure so that the whole process could be captured from a single canvass.

Consider including a diagram showing the linkages between qualitative and quantitative components and how it fits into the principles of the mixed-method design used for this study. What was done to bring out what, and the actual outcome?

Have a closer look at lines 45-47 on page 2 (anything missing?).

Page 4; Lines 84-84: What is the percentage deficiency?

Results & Discussion:

To improve readability and comprehension, in tables, consider writing the norms for each category of staff and against that write the actual number of staff in place, followed by number of vacant posts and percentage deficiency.

What are the reasons for health worker shortages? What are the underlying factors?

Student intake for these professional courses that produce these personnel? Government / private break-up? Share of out-migration, and in-migration, especially that of nurses; pay in neighbouring countries vis-a-vis migration?

What were the major themes that emerged from content analysis?

References:

Try to include more references to support rationale and findings.

What was the search strategy?

If organization is the author, write the name completely in the same order it is written in the source document; do not reformat it into first name, followed by middle and last names (e.g., reference numbers 4 and 14); do not abbreviate the name. (First name, followed by initials is recommended for individual authors, not for organizations).

Check names of authors and initials under the list of references.

General - formatting, etc.

Rephrase some of the sentences.

e.g, Page 11; Lines 260-72: Either write percent in ...... (name of facility), OR write name of facility followed by percent within brackets.

Check grammar: e.g., 'data was'.

Spell check the document - using MS Word AND proof-read manually, e.g., Page 19; Line 382: 'per-urban'.

Reviewer #4: General Comments

In the abstract and introduction section, Kenya’s challenges with the availability of HRH is presented but it is not clear how this affects the attainment of Universal Health Coverage (UHC).

In lines 71-73 on page 3, the following sentence is hanging: “In 2017, building on the Road map for scaling up HRH for improved health service delivery in the African Region 2012–2025 that was adopted by the Sixty-second session of the Regional Committee.”

In lines 119-123 on page 5. The interviewees are listed as the study population! I think this is not correct. I thought the study population should be the total number of health facilities in the study areas.

In lines 137-138 on page 6, it is stated that “Informants were purposively selected from various categories. A total of 36 FGDs and 30 IDIs per county were targeted.” What informed the targeting? How many people were in each FGD? Could you describe the FGD process in terms of recruitment, discussion process, number of iterations, etc.

In lines 140-143 on page 6. The differences between the outcomes and exposures are not clear. The sentences defining these terms seem to be the same to me.

In lines 144-145 on page 6, the definition for the predictors is provided. However, these are not used to interpret the results and in the discussion. The authors could use the predictors to explain the factors that influence the availability and distribution of HRH and the implementation of UHC in Kenya.

On page 8 (lines 188-208), the section on ethical consideration is very long. This could be cut to three short sentences.

In the results section, the information provided from lines 212-234 (page 9) could be cut and taken to the methodology section.

Please add p-values in the tables where necessary.

Table 4 could be summarized. Too many details are provided most of which could be summarized in a short table. The full table could be part of the supplementary documents.

Results from the quantitative and qualitative components of the study are presented separately. They could be presented together and/or alongside to improve interpretation.

Overall, the discussion is very long and there is inadequate triangulation of the results from the study with other studies. The authors must ensure that the results are linked to the objectives of the study, sufficiently interpreted and triangulated. In most cases, the previous studies and examples that have been used in the discussion section to interpret the results are not linked to actual findings from the study.

The conclusion as at the end of the article is very too long and not sufficiently linked to the objectives of the study. The authors seem to be presenting the results again rather than concluding. The authors need to prepare a conclusion with only 2-3 sentences linked to the objectives of the study. The recommendations and areas for future research also need to be provided and linked to the conclusion of the study.

Limitations

About three limitations have been provided but the authors have not explained how these limitations were addressed or mitigated in the study. The authors must address this issue. Secondly, the study didn’t look at the effect or impact of non-availability of HRH in the achievement on UHC. On the other hand, increased availability of health workers (including optimal distribution and skills-mix) is a necessary but not the absolute condition for success or progression to UHC. What if the conditions of service are poor and the salaries are low? The health workers could be unproductive? What if the health workers have low skills? What if the other health systems inputs like drugs, vaccines, medical equipment, etc are not available?

The analysis also misses a budget and futuristic financing landscape analysis to determine if meeting the staffing norms is feasible in view of the financial resources that the Kenyan government has now and in future. Lastly, the study also misses an analysis of the HRH training outputs and health workers currently not employed to fill the set staffing norms. This analysis could complement the analysis on the staffing gaps vis-à-vis the availability of labor to fill the gaps, and government’s financing ability/capacity to employ.

Reviewer #5: (No Response)

7. PLOS authors have the option to publish the peer review history of their article (what does this mean?). If published, this will include your full peer review and any attached files.

Reviewer #3: No

Reviewer #4: **Yes: **Collins Chansa

Reviewer #5: No

---

## [Decision Letter · Decision Letter 2]

11 Dec 2023

PONE-D-23-11190R2Health systems’ capacity in availability of human resource for health towards implementation of Universal Health Coverage in KenyaPLOS ONE

Dear Dr. Ahmed,

Thank you for submitting your manuscript to PLOS ONE. After careful consideration, we feel that it has merit but does not fully meet PLOS ONE’s publication criteria as it currently stands. Therefore, we invite you to submit a revised version of the manuscript that addresses the points raised during the review process, particularly minor revision as per the comments from one of the reviewers(described below).

We look forward to receiving your revised manuscript.

Kind regards,

Philipos Petros Gile, MA

Academic Editor

PLOS ONE

Journal Requirements:

Reviewers' comments:

Reviewer's Responses to Questions

**Comments to the Author**

1. If the authors have adequately addressed your comments raised in a previous round of review and you feel that this manuscript is now acceptable for publication, you may indicate that here to bypass the “Comments to the Author” section, enter your conflict of interest statement in the “Confidential to Editor” section, and submit your "Accept" recommendation.

Reviewer #3: (No Response)

Reviewer #5: All comments have been addressed

2. Is the manuscript technically sound, and do the data support the conclusions?

Reviewer #3: Yes

Reviewer #5: Yes

3. Has the statistical analysis been performed appropriately and rigorously? 

Reviewer #3: I Don't Know

Reviewer #5: Yes

4. Have the authors made all data underlying the findings in their manuscript fully available?

Reviewer #3: Yes

Reviewer #5: Yes

5. Is the manuscript presented in an intelligible fashion and written in standard English?

Reviewer #3: Yes

Reviewer #5: Yes

6. Review Comments to the Author

Reviewer #3: Some minor issues to be sorted out:

1. P 3; L 81-82: Reference for this statement?

2. P 4; L 92: Missing word - 'number'?

3. P 6; L 151: Replace 'interviewed' with 'conducted'.

4. P 11; L 269: '.... in rural areas have meet staffing norms ........? 

5. P 14; L 301: Check heading - 'carders'

6. P 14; L 311: Check heading - 'RH'

7. P 16; L 359-360: 'The overall HRH availability index of 17.2 percent' - against what?

8. P 17; L 362: Insert period (.) after 'healthcare services'.

9. P 17; L 365-366: 'WHO recommended minimum'. - Mention what is the WHO recommendation; and what is the percentage of deficiency vis-a-vis the recommendation.

10. P 17; L 372: Comparable HRH available indexes - what are they? Menton or give reference to 'Table' if referred to in the Table.

11. P 18; L 409: '.... had the necessary' - Look for missing word(s).

12. P 22-23: References - (a) Author names remain incomplete or incorrect. (b) Inappropriate way of abbreviating organizational author (e.g., WHO).

GENERAL:

Check spelling, grammar; singular-plural  (e.g., data 'was')Check whether past tense is used wherever required.

Make sure that key points / wordings  mentioned in Line numbers 119-120 and the wordings mentioned under 'Results' are in alignment; and that whatever the authors have committed vide Line 119-120 are reported under relevant headings.

Make sure that all citations in the text are acknowledged and assigned a number and that they are listed under the heading 'References'.

Reviewer #5: (No Response)

7. PLOS authors have the option to publish the peer review history of their article (what does this mean?). If published, this will include your full peer review and any attached files.

Reviewer #3: No

Reviewer #5: **Yes: **Valerian Mwenda

---

## [Decision Letter · Decision Letter 3]

4 Jan 2024

Health systems’ capacity in availability of human resource for health towards implementation of Universal Health Coverage in Kenya

PONE-D-23-11190R3

Dear Author,

We’re pleased to inform you that your manuscript has been judged scientifically suitable for publication and will be formally accepted for publication once it meets all outstanding technical requirements and minor comments from reviewers.

Kind regards,

Philipos Petros Gile, MA

Academic Editor

PLOS ONE

Additional Editor Comments (optional):

Reviewers' comments:

Reviewer's Responses to Questions

**Comments to the Author**

1. If the authors have adequately addressed your comments raised in a previous round of review and you feel that this manuscript is now acceptable for publication, you may indicate that here to bypass the “Comments to the Author” section, enter your conflict of interest statement in the “Confidential to Editor” section, and submit your "Accept" recommendation.

Reviewer #3: (No Response)

2. Is the manuscript technically sound, and do the data support the conclusions?

Reviewer #3: Yes

3. Has the statistical analysis been performed appropriately and rigorously? 

Reviewer #3: I Don't Know

4. Have the authors made all data underlying the findings in their manuscript fully available?

Reviewer #3: Yes

5. Is the manuscript presented in an intelligible fashion and written in standard English?

Reviewer #3: Yes

6. Review Comments to the Author

Reviewer #3: The authors have addressed all queries except the last one - References. These are yet to be formatted correctly.

7. PLOS authors have the option to publish the peer review history of their article (what does this mean?). If published, this will include your full peer review and any attached files.

Reviewer #3: No

---

## [Editor Report · Acceptance letter]

22 Jan 2024

PONE-D-23-11190R3 

PLOS ONE

Dear Dr. Ahmed, 

I'm pleased to inform you that your manuscript has been deemed suitable for publication in PLOS ONE. Congratulations! Your manuscript is now being handed over to our production team.

Kind regards, 

on behalf of

Dr. Philipos Petros Gile 

Academic Editor

PLOS ONE